# Transcriptomic Landscape and Functional Characterization of Human Induced Pluripotent Stem Cell-Derived Limbal Epithelial Progenitor Cells

**DOI:** 10.3390/cells11233752

**Published:** 2022-11-24

**Authors:** Naresh Polisetti, Julian Rapp, Paula Liang, Viviane Dettmer-Monaco, Felicitas Bucher, Jan Pruszak, Ursula Schlötzer-Schrehardt, Toni Cathomen, Günther Schlunck, Thomas Reinhard

**Affiliations:** 1Eye Center, Medical Center, Faculty of Medicine, University of Freiburg, Killianstrasse 5, 79106 Freiburg, Germany; 2Institute for Transfusion Medicine and Gene Therapy, Medical Center, University of Freiburg, 79106 Freiburg, Germany; 3Freiburg iPS Core, Center for Chronic Immunodeficiency, Medical Center, University of Freiburg, 79106 Freiburg, Germany; 4Department of Ophthalmology, University Hospital Erlangen, Friedrich-Alexander-University of Erlangen-Nürnberg, Schwabachanlage 6, 91054 Erlangen, Germany

**Keywords:** induced pluripotent stem cells, limbal stem cells, limbal niche cells, melanocytes, limbal epithelial progenitor cells, corneal tissue engineering, RNA-sequencing, immuneresponse, anigogenesis, decellularized limbal scaffolds

## Abstract

Limbal stem cell deficiency (LSCD) is a complex, multifactorial disease affecting limbal epithelial progenitor cells (LEPC), which are essential for maintaining corneal stability and transparency. Human induced pluripotent stem cell-derived (hiPSC-) LEPC are a promising cell source for the treatment of LSCD. However, their similarity to native tissue-derived (T-) LEPC and their functional characterization has not been studied in detail. Here, we show that hiPSC-LEPC and T-LEPC have rather similar gene expression patterns, colony-forming ability, wound-healing capacity, and melanosome uptake. In addition, hiPSC-LEPC exhibited lower immunogenicity and reduced the proliferation of peripheral blood mononuclear cells compared with T-LEPC. Similarly, the hiPSC-LEPC secretome reduced the proliferation of vascular endothelial cells more than the T-LEPC secretome. Moreover, hiPSC-LEPC successfully repopulated decellularized human corneolimbal (DHC/L) scaffolds with multilayered epithelium, while basal deposition of fibrillary material was observed. These findings suggest that hiPSC-LEPC exhibited functional properties close to native LEPC and that hiPSC-LEPC-DHC/L scaffolds might be feasible for transplantation in patients suffering from LSCD in the future. Although hiPSC-LEPC-based stem cell therapy is promising, the current study also revealed new challenges, such as abnormal extracellular matrix deposition, that need to be overcome before hiPSC-LEPC-based stem cell therapies are viable.

## 1. Introduction

Human limbal stem/progenitor cells (LEPC), located at the basal layer of the corneoscleral limbus, are responsible for homeostasis of the corneal epithelium [1]. Pathological conditions such as thermal or chemical burns can lead to limbal stem cell deficiency (LSCD) resulting in conjunctivalization of the corneal surface, corneal neovascularization, recurrent and persistent epithelial defects, chronic inflammation, scarring, ulceration of the cornea, and finally loss of vision up to blindness [2,3]. Various medical treatment options were reported to restore the limbal microenvironment and optimize the ocular surface including transplantation of corneal limbal tissue, ex vivo expanded LEPC or oral mucosal epithelial cells (OMEC) [3,4,5]. Although promising outcomes were achieved in many cases, certain drawbacks persist, given that (i) autologous LEPC transplantation puts the healthy cornea of the contralateral donor eye at risk, (ii) allogenic transplantation is limited by a shortage of donors or immunological incompatibility, and (iii) OMECs exert only a weak anti-angiogenic function, which leads to neoangiogenesis following transplantation [6,7,8]. Therefore, the use of LEPC derived from human-induced pluripotent stem cells (hiPSC-LEPC) seems a promising alternative approach, especially in bilateral LSCD. 

Several protocols to differentiate hiPSC into corneal epithelial lineages have been reported [9,10,11]. Many protocols draw from the understanding of ectoderm development and suggest modulating distinct signaling pathways, e.g., by using small molecule inhibitors to block transforming growth factor beta and Wnt signaling followed by activation of fibroblast growth factor or bone morphogenetic protein pathways [10,12]. A different approach promotes autogenic eye-like differentiation of human iPSC in self-forming ectodermal autonomous multi-zones (SEAM) giving rise to several ocular cell types [13] including corneal epithelial cells, which enabled functional recovery in an animal model of LSCD [13]. The gene expression profile of hiPSC-derived corneal epithelial cells has been analyzed in comparison with native differentiated limbal epithelial cells (paired box (PAX)6^+^ cytokeratin(CK)12^+^) [13] but not with LEPC (PAX6^+^CK12^−^), which are present in the basal layer of limbal epithelium and can be isolated by flow sorting (P-cadherin^+^CD90^−^CD117^−^) [14]. Moreover, it is important to evaluate the angiogenic and immunogenic potential of hiPSC-LEPCs before applying them clinically, as many factors influence the immunological and angiogenic properties of hiPSC-LEPC, such as long differentiation times and culture medium composition [15]. In terms of immune-related gene expression, T-cell proliferation, and natural killer cell lysis, limbal stem cells derived from embryonic stem cells (ESC) and corneal epithelial progenitor cells derived from iPSC were less immunogenic in vitro than native limbal stem cells or corneal epithelial progenitor cells [16,17]. Additionally, when compared to native limbal stem cells, ESC derived limbal stem cells have shown less infiltration of immune cells after being transplanted in rabbits [16]. However, a detailed study of hiPSC-LEPC immunogenicity and angiogenic potential has not been reported.

The long-term success of limbal stem cell transplantation is largely determined by the maintenance and survival of LEPC within the graft, which restores a functional stem cell pool [18]. In contrast, standard culture techniques ignore the subtleties of the niche microenvironment, which is primarily composed of extracellular matrix (ECM) and niche cells and regulates stem cell homeostasis in vivo [19]. As a result, current research focuses on identifying limbal niche components and developing appropriate biomimetic scaffolds to replicate the limbal niche in vitro to a certain extent [20,21]. Decellularized human corneal/limbal (DHC/L) scaffolds have emerged for corneal epithelial cell delivery since they not only provide the native ECM composition, but also the three-dimensional corneal/limbal architecture [22,23]. However, the repopulation of human DHC/L scaffolds and reconstruction of the limbal stem cell niche using hiPSC-LEPC has not been explored. Furthermore, no studies have been performed to examine the interactions of hiPSC-LEPC with limbal melanocytes (LM), a niche cell component which protects LEPC from UV damage by transferring melanin granules, control LEPC homeostasis, and have effective immunomodulatory and anti-angiogenic properties [22,24,25,26,27,28]. 

In the present study, we characterized and tested functional properties of hiPSC-LEPC generated using a previously established clinically applicable protocol in vitro [13]. For iPSC-differentiated cells, we used the terminology LEPC since these cells showed the phenotypic signature of PAX6^+^CK3/12^−/low^, which are generally found in the basal layer of the limbus. The phenotypic similarity of hiPSC-LEPC to T-LEPC was investigated by transcriptomic analysis using next-generation sequencing. We also explored the additional functional role of hiPSC-LEPC in immune regulation and angiogenesis. Moreover, we examined the ability of hiPSC-LEPC to repopulate DHC/L scaffolds, which could serve as carriers for cell transplantation in ocular surface reconstruction. 

## 2. Material and Methods

### 2.1. Tissue and Cells

KYOUDXR0109B Human Induced Pluripotent Stem Cells (hiPSCs) [201B7] (ATCC^®^ ACS1023™) cell line 201B7 were purchased from the American Type Culture Collection (ATCC) and cultivated in a feeder-free system in the presence of laminin (LN)-511 E8 fragments (iMATRIX-511, Tokyo, Japan) and animal component-free culture medium (TeSR-E8; Stem cell technologies, Cologne, Germany).

Organ cultured human corneoscleral tissues (n = 25) were obtained from the LIONS Cornea Bank Baden-Württemberg, Eye Center, University Medical Center Freiburg. Informed consent for research use of tissue remnant after transplant excision had been given by the donors or their next of kin. To isolate primary limbal cells, organ cultured corneoscleral rims remaining after penetrating keratoplasty were used (n = 25, Appendix A). The study was approved by the institutional review board of the University of Freiburg (25/20) and followed the tenets of the Declaration of Helsinki. Human umbilical vein endothelial cells (HUVEC) were purchased from Lonza (Basel, Switzerland) and were cultured in endothelial growth medium (EGM) containing endothelial basal medium and endothelial cell growth supplement (Lonza, Basel, Switzerland). 

All the experiments were done with multiple batches of LEPC independently derived from 201B7 cell lines. 

### 2.2. Differentiation of hiPSC Using the SEAM Method

The differentiation of hiPSC was performed according to a previously reported protocol [13] with few modifications. First, hiPSCs were dissociated using an enzyme-free reagent (ReleSR; StemCell Technologies, Cambridge, MA, USA) as per manufacturer’s instructions. The clusters obtained were seeded on laminin (LN)-511 E8 (i-Matrix, Nippi, Tokyo, Japan)-coated dishes at 10–20 clusters cm^−2^ and cultured in TeSR-E8 medium for 5–8 days. The culture medium was then changed to differentiation medium (DM; GMEM (Gibco, Thermofisher Scientific, Dreieich, Germany) supplemented with 10% knockout serum replacement (KSR; Life Technologies, Karlsruhe, Germany), 1 mM sodium pyruvate (Life Technologies, Karlsruhe, Germany), 0.1 mM non-essential amino acids (Life Technologies, Karlsruhe, Germany), 2 mM l-glutamine (Life Technologies, Karlsruhe, Germany), 1% penicillin-streptomycin solution (Life Technologies, Karlsruhe, Germany) and 55 μM 2-mercaptoethanol (Life Technologies, Karlsruhe, Germany). After four weeks, the medium was changed to corneal differentiation medium (CDM containing DM and Cnt-PR (1:1) (CELLnTEC Advanced Cell Systems, Bern, Switzerland), 20 ng/mL keratinocyte growth factor (KGF) (Wako, Neuss, Germany) 10 μM Y-27632 (Wako, Neuss, Germany) and 1% penicillin-streptomycin solution). After four weeks of culture in CDM, the medium was changed to corneal epithelium maintenance medium (CEM; DMEM/F12 (2:1) (Life Technologies, Karlsruhe, Germany) containing 2% B27 supplement (Life Technologies, Karlsruhe, Germany), 1% penicillin-streptomycin solution, 20 ng/mL KGF and 10 μM Y-27632 for six to seven weeks.

### 2.3. Cell Sorting

Differentiated hiPSC in CEM were dissociated using Accutase (Life Technologies, Karlsruhe, Germany) and resuspended in ice-cold staining buffer (Keratinocyte serum-free medium supplemented with 2% fetal bovine serum (FBS) and IX insulin transferrin selenium solution (Life Technologies, Karlsruhe, Germany). The harvested cells were filtered with a cell strainer (40 µm, BD Biosciences, Heidelberg, Germany) and flow sorting was performed according to our previously reported method [14,29]. Briefly, single-cell suspensions were incubated with conjugated antibodies (Appendix A) on ice for 1 h. After three washes, cells were resuspended in ice-cold PBS, and flow cytometry was performed on a FACSCanto II (BD Biosciences, Heidelberg, Germany ) using FACS Diva Software (BD FACSDiva 8.0.1, BD Pharmingen; BD Biosciences, Heidelberg, Germany). Appropriate compensation was carried out and post-acquisition analysis was performed using FlowJo software (FlowJo 10.2, Tree Star Inc., Ashland, OR, USA) (n = 13).

Sorted hiPSC-derived limbal epithelial cells (CD200^−^ITGB4(integrin beta-4)^+^SSEA-4(stage-specific embryonic antigen-4)^+^; considered as P0 and referred to as hiPSC-LEPC) were seeded on 3T3 fibroblasts as mentioned below for colony forming assays or seeded into LN511E8 coated (0.5 µg/cm^2^) 6-well plates and cultured in CEM medium. 

T-LEPC (CD90^−^CD117^−^P-cadherin^+^) and LM (CD90^−^CD117^+^P-cadherin^+^) from cadaveric limbal tissues were isolated and cultured as described previously [14]. 

### 2.4. Total RNA Extraction

Total RNA extraction from both hiPSC-LEPC (P1, n = 7) and LEPC (P1, n = 4) were performed as described previously [30] using RNeasy Plus Micro Kit (Qiagen, Hilden, Germany). 

### 2.5. RNA Sequencing

RNA sequencing was carried out as described previously [30]. The SMARTer Ultra Low Input RNA Kit for Sequencing v4 (Clontech Laboratories, Inc., Mountain View, CA, USA) was used to generate first strand cDNA from 1 ng total-RNA. Double-stranded cDNA was amplified by LD PCR (10 cycles) and purified via magnetic bead clean-up. Library preparation was carried out as described in the Illumina Nextera XT Sample Preparation Guide (Illumina, Inc., San Diego, CA, USA). 150 pg of input cDNA were tagmented (tagged and fragmented) by the Nextera XT transposome. The products were purified and amplified via a limited-cycle PCR program to generate multiplexed sequencing libraries. For the PCR step 1:5 dilutions of index 1 (i7) and index 2 (i5) primers were used. The libraries were quantified using the KAPA Library Quantification Kit—Illumina/ABI Prism User Guide (Roche Sequencing Solutions, Inc., Pleasanton, CA, USA). Equimolar amounts of each library were sequenced on a NextSeq 500 instrument controlled by the NextSeq Control Software (NCS) v2.2.0, using a 75 Cycles High Output Kit with the dual index, single-read (SR) run parameters. Image analysis and base calling were done by the Real Time Analysis Software (RTA) v2.4.11. The resulting .bcl files were converted into .fastq files with the bcl2fastq v2.18 software. 

RNA extraction, library preparation and RNAseq was performed at the Genomics Core Facility “KFB—Center of Excellence for Fluorescent Bioanalytics” (University of Regensburg, Regensburg, Germany).

The sequence data are available at the Gene Expression Omnibus database under accession number PRJNA875057. The password is available from the corresponding author upon request.

### 2.6. Bioinformatics 

Bioinformatics analysis was carried out as described previously [30]. Raw data were uploaded to the galaxy.eu platform (https://usegalaxy.eu, accessed on 15 April 2022) [31] and quality assessed by FastQC software [32] (Galaxy Version 0.73). Adapters were clipped by Trim Galore! (Galaxy version 0.6.7) and reads mapped to a human reference genome (GRCh38) provided by GENCODE (https://www.gencodegenes.org, downloaded on 14 April 2022) and the corresponding gene annotation file (downloaded on 14 April 2022) using the STAR aligner [33] (Galaxy Version 2.7.8a) for single-end reads in standard settings. Aligned reads were assigned to features by featureCounts [34] (Galaxy Version 2.0.1) using the mentioned gene annotation file from GENCODE and counts imported in R 4.0.2 (https://www.rproject.org, accessed on 15 April 2022). Detection of differentially expressed genes was conducted by the DESeq2 package [34] and ENSEMBL gene ID linked to HGNC system by biomaRt [35]. Differentially expressed genes (DEG) were defined by padj < 0.05 and absolute log2-fold change > 2. Gene Ontology (GO) enrichment analysis was conducted by using the clusterProfiler package [36]. Gene Set Enrichment Analysis (GSEA) was performed with fgsea [37] using gene sets from MSigDB database (http://www.gseamsigdb.org/gsea/msigdb, downloaded 15 April 2021) by sorting genes according to their shrunken log2-fold change generated by DESeq2 in normal settings and excluding genes with <10 counts. Similarity between groups was determined by ranking each gene according to their normalized expression calculated by DESeq2 in percentiles in the respective group. The difference between genes across groups was defined by subtracting the percentiles of each gene from each other and further visualized by color. Overall similarity was measured by calculating the Pearson correlation coefficient. Genes with a base count smaller then 10 were excluded. The ggplot2 [36] and ComplexHeatmaps [38] packages were used for data visualization. 

### 2.7. CFU-E Assay

Clonal expansion of hiPSC-LEPC (P0, 300 cells/cm [2]) were studied on feeder layers using mitomycin C-treated 3T3 fibroblasts as described previously [14] (n = 4). 

### 2.8. Flow Cytometry 

Flow cytometry was carried out as described previously [29]. Briefly, single-cell suspensions (0.5–1 × 10^6^ cells) were incubated with fluorochrome-conjugated antibodies and respective isotype controls (Appendix A). After three washes, cells were resuspended in ice-cold PBS, and flow cytometry was performed on a FACSCanto II (BD Biosciences) by using FACS Diva Software (BD FACSDiva 8.0.1). A total of 10,000 events were acquired. A Post-acquisition analysis was conducted using the FlowJo software (FlowJo 10.2, Tree Star Inc. Ashland, OR, USA).

### 2.9. Co-Culture of LMs with hiPSC-LEPC 

To study the possible interactions of LM and hiPSC-LEPC, the co-culture experiments were performed as demonstrated in our earlier publication [14] (n = 3). 

### 2.10. Wound Healing Assays

For wound healing assays, hiPSC-LEPC (P1) were seeded at 25,000/well in 96-well IncuCyte ImageLock Plates coated with laminin-511E8 as described above. T-LEPC (P1) was used as a control. After confluence, the scratch was then made using the 96-pin wound maker (Essen BioScience, Ann Arbor, MI, USA), and wells were washed two times with PBS to remove floating cells. Immediately after the following washing, CEM media was added. Images of the wounded areas were taken automatically every hour for 24 h using the Inucucyte Live-Cell Analysis platform, and the data were analyzed by the integrated metric Relative Wound density part of the live content imaging system IncuCyte HD (Incucyte^®^ Scratch Wound Analysis Software, Version 20213.2.0.0, Essen BioScience Inc., Sartorius, Goettingen, Germany) (n = 4). 

### 2.11. Angiogenesis-Related Assays

Conditioned media: To investigate the effect of conditioned media (CM) on angiogenesis (pro or anti), HUVEC up to passage 6 were used. For the preparation of CM, equal numbers of LEPC and hiPSC-LEPC were cultivated overnight on tissue culture plates in their respective medium as mentioned above. On the next day, the cell-specific culture media were replaced by Endothelial cell growth basal medium-2 (EBM-2; Lonza) containing 0.5% FCS. After 48 h, CM was collected and centrifuged at 1000 g for 10 min at 4 °C. The amount of CM used was calculated relative to 5 × 10^5^ cells/mL.

Proliferation assay: The effect of CM on HUVEC proliferation was quantified using the cell proliferation ELISA BrdU colorimetric Assay kit (Roche Diagnostics, Mannheim, Germany). Cells were seeded at a density of 2500 cells/well in 96-well plates in EGM. After 6 hrs of seeding, the cells were starved overnight in EBM-2 containing 2% FCS. In the following day (day 1), the cells were stimulated with CM either undiluted or diluted (1:1 with EBM-2 + 2% FCS). Cells were cultured in CM for 72 h and labeled with BrdU according to manufacturers’ instructions. Absorbance was measured at 450 nm using a Spark microplate reader (TECAN). Cells treated with human vascular endothelial growth factor (hVEGF, 25 ng/mL, Peprotech, Rocky Hill, NJ, USA) only served as a positive control, and cells treated with EBM-2 served as negative controls. The fold change values were calculated as OD of the CM/OD of control (n = 4)

Incucyte Migration assay: For endothelial cell migration assays, HUVEC were seeded at 25,000/well in 96-well IncuCyte ImageLock Plates. After 6 h of seeding, growth media were replaced with EBM-2 containing 2% FCS and starved overnight. The scratch was then made using the 96-pin wound maker (Essen BioScience, Ann Arbor, MI, USA), and wells were washed two times with EBM media to remove floating cells. Immediately after the following washing, the media were replaced with CM of T-LEPC and hiPSC-CEC in an undiluted and diluted setup with EBM-2 (1:1 ratio). Positive and negative control cells were treated as described above. Images of the wounded areas and the data were analyzed as described above (n = 7).

Endothelial spheriod sprouting assay: The endothelial spheroid sprouting assay was also performed as previously published protocol [39]. Briefly, endothelial spheroids were formed in a hanging drop consisting of 200,000 HUVECs resuspended in 8 mL EGM and 2 mL methylcellulose (#M0512; St. Louis, MO, USA.). Spheroids were harvested the following day and poured into a mixture of 2 mL collagen I (Collagen I, Rat Tail, Corning, New York, NY, USA.) containing 2% FBS, 400 µL EBM and 2 mL Methocel in 24-well plates with a volume of each 0.5 mL Spheroid-containing gel which solidified at 37 °C for 30 min and was then stimulated with CM of T-LEPC and hiPSC-CEC in undiluted and diluted mixture with EBM-2 + 2% FCS in a 1:1 ratio. Positive and negative control cells were treated as described above. Images of single spheroids were taken using an inverted light microscope (Zeiss Axio Vert, Zeiss, Oberkochen, Germany). Total sprout length was determined by using a self-programmed ImageJ (ImageJ Fiji, 2.0.0-rc-69/1.52i, https://imagej.nih.gov/ij/, downloaded on 20 April 2021) macro where sprouts must be manually labeled using the “straight-line” tool. The length of all sprouts of one spheroid was added up and the average total length per spheroid calculated for each condition. As a final readout, the data were normalized to controls to form the relative sprouting length (RSL) (n = 7). 

### 2.12. Immunomodulatory Assays

IFN-γ treatment: To stimulate the inflammatory influence on the immunogenicity, hiPSC-LEPC (P1) and T-LEPC (P1) were treated with interferon-gamma (IFN-γ) (Sigma-Aldrich, St. Louis, MO, USA) at a concentration of 10 ng/mL for 24 h [24]; Untreated T-LEPC and hiPSC-LEPC served as control. Flow cytometry was performed as described above (n = 4). 

Co-culture of PBMCs with hiPSC-LEPC: Primary blood mononuclear cells (PBMCs) from three HLA-unmatched healthy donors were extracted from Leukocyte Reduction System (LRS) chambers containing white blood cells (WBCs) that were retained after routine donor plateletpheresis. LRS chambers were kindly provided by the Blood Donation Center of the Medical Center in Freiburg after informed consent of the anonymized PBMC donors. PBMCs were obtained by Bicoll density gradient centrifugation according to manufacturer’s instructions and cultured as described previously [40]. For activation, monocytes were depleted for 4 h by letting them attach naturally to the bottom of the plates. Lymphocyte-enriched cells from the supernatant were then incubated for three days in the presence of ImmunoCult™ Human CD3/CD28/CD2 T Cell Activator (Stem Cell) at a concentration of 5 µL/1 × 10^6^ cells/mL. 

For co-cultures, hiPSC-LEPC (P1) or T-LEPC (P1) were respectively seeded at a density of 100,000, 50,000, 25,000 and 10,000 cells/well of a flat-bottom 96 well-plate. After 24 h seeding, some wells were stimulated with interferon gamma (IFN-γ, 10 ng/mL) for 24 h. After 48 h of seeding, either non-activated and activated PBMCs labeled with carboxyfluorescein succinimidyl ester (CFSE) as per manufacturer instructions, they were cultured (1 × 10^5^ cells/well) alone or, alternatively, the cells were added to the wells with pre-seeded hiPSC-LEPC or T-LEPC and cultured for 72 h (n = 4). 

### 2.13. Repopulation of Decellularized Corneal/Limbal Scaffolds with hiPSC-LEPC

The decellularization of corneoscleral rims and recellularization of decellularized scaffolds were performed according to our previously reported method [23]. The recellularized scaffolds were fixed for immunohistochemistry or light and electron microscopy as described below.

Live/dead viability/cytotoxicity kit (MP 03,224, Molecular Probes, Eugene, OR, USA) was used to visualize live and dead cells in repopulated DHC/L as described previoulsy [22] (n = 4). 

### 2.14. Histology and Immunohistochemistry—Paraffin

For routine histology, the scaffolds were fixed in 4% paraformaldehyde for 30 min and embedded in paraffin. The 5 µm thick sections were cut and stained with H&E (hematoxylin (Haematoxylin Gill III, Surgipath, Leica, Wetzlar, Germany) and eosin Y (Surgipath, Leica, Germany) as described previously [22]. Immunohistochemistry was performed as previously described [41] (n = 3). The list of antibodies is provided in Appendix A. 

### 2.15. Immunohistochemistry of Frozen Sections and Immunocytochemistry

Immunostaining of frozen sections and cultured cells was performed as described in our earlier publication [23] (n = 3). 

### 2.16. Transmission Electron Microscopy

For transmission electron microscopy (TEM), tissue specimens were processed as described previously [23] (n = 3). 

### 2.17. Statistical Analysis

All assays were performed by at least four independent experiments. Statistical analyses was performed using the GraphPad Instat statistical package for windows (Version 6.0; Graphpad Software Inc., La Jolla, CA, USA) and all the data were presented as mean ± standard error of the mean (SEM). The statistical significance (*p* value < 0.05) was determined with Wilcoxon signed-rank test or Mann–Whitney U test as appropriate. Statistical analysis in MLR were Mann–Whitney U tests. 

## 3. Results 

### 3.1. Isolation and Characterization of hiPSC-LEPC

Human iPSC (Figure 1A(i)) were differentiated towards a limbal epithelial phenotype by SEAM formation (Figure 1A(ii)) [13]. After 12–14 w of differentiation (Figure 1A(iii)), cells were treated with Accutase and the obtained single cell suspension was subjected to flow sorting. The single cell suspensions were gated to select cells of interest and to enrich single cells followed by dead cell exclusion using 4′,6-diamidino-2-phenylindole (DAPI) as described [29]. Then the gates were set based on isotype controls to select CD200^−^ cells initially and followed by a subset selection of CD200^−^ ITGB4^+^SSEA4^+^ cells (Figure 2B) [42]. Flow sorting analysis revealed that 7–11% of cells were CD200^−^SSE4^+^ITGB4^+^ (referred as hiPSC-LEPC; Figure 2B). The cultured hiPSC-LEPC (P0) exhibited a cuboidal epithelial phenotype and grew like colonies (Figure 1C). Immunostaining of cultured hiPSC-LEPC (P1) showed the expression of PAX6 (99 ± 2/100), p63 (97 ± 2/100), CK14 (96 ± 3/100), CK19 (98 ± 3/100), and CK15 (36 ± 5/100), CK3 (8 ± 4/100) and the proliferation marker Ki-67 (54 ± 7/100) (Figure 1D). The hiPSC-LEPC were also characterized by Western blotting for limbal epithelial markers (Appendix A). The expression of E-cadherin and P-cadherin, cell-cell adhesion molecules which have been shown to mediate stem cell niche interactions at the limbal stem cell niche [24], were examined by immunostaining. E-cadherin and P-cadherin (red) were diffusely stained in hiPSC- LEPC (P1) cultured in 0.08 mM Ca^2+^ and enriched at cell-cell junctions in medium containing 1.2 mM Ca^2+^ (Figure 2A). We also observed gaps between the cells at 1.2 mM Ca^2+^ (Figure 2A, dotted circles).

We measured the colony-forming capabilities to address a key property of stem/progenitor cells. Human iPSC-LEPC formed colonies with smooth borders (Figure 2B(i), dotted line demarcates the hiPSC-LEPC colony) and a colony-forming efficiency of 2.6 ± 0.3% (Figure 2B(ii)), suggesting the presence of clonogenic epithelial progenitor cells. Immunocytochemical staining of hiPSC-LEPC colonies (pan-CK, green) on 3T3 feeder layers (vimentin, red) showed that 3T3 fibroblasts concentrated around the edges of the colonies (Figure 2C). Double immunostaining of hiPSC-LEPC clones showed membranous staining of E- (red) and P-cadherins (green; Figure 2C). The limbal epithelial progenitor marker CK15 (green) and the proliferation marker Ki-67 (red) were detected mostly at the edges of the colonies. The corneal differentiation marker CK3 (green) was found in many cells of a colony and the progenitor cell marker p63 (red) was present in almost all cells of the clone (Figure 2C). To assess the migration potential of hiPSC-LEPC during wound healing, a scratch was made on a confluent hiPSC-LEPC or LEPC monolayer using the 96-pin wound maker. Wound closure rates of both hiPSC-LEPC (Phase contrast, Figure 2D(i,ii)) and T-LEPC were similar at any given time point (Figure 2D(iii))

To study the interaction of hiPSC-LEPC with LM, hiPSC-LEPC and LM were co-cultured. Co-cultures of LM and hiPSC-LEPC showed infiltration of LM in the hiPSC-LEPC colonies (Figure 2E(i), stars depict the melanocytes and their processes, similar to observations made in T-LEPC [26]). Upon contact with surrounding hiPSC-LEPCs, LM formed long, dendritic cell processes which ensheathed the hiPSC-LEPC (Figure 2E(i)) as seen in the limbal niche in situ [26]. Double immunostaining of co-cultures showed the accumulation of pigment globules (arrow; HMB45, green) in the perinuclear cytoplasm of hiPSC-LEPC (E-cadherin, red, Figure 2E(ii,iii)). 

### 3.2. Transcriptional Profiling of hiPSC-LEPC

To further determine the resemblance of hiPSC-LEPC and T-LEPC, we performed a transcriptome analysis using RNA-sequencing. The hiPSC-LEPCs from seven independent experiments and T-LEPCs from four independent experiments (comprising 4–6 biological donors each) were subjected to total RNA-sequencing analysis. Principal component analysis (PCA) shows that T-LEPC cluster closely together, whereas hiPSC-LEPC show more variability between samples (Figure 3A). To assess the level of similarity between hiPSC-LEPC and T-LEPC, we performed a correlation analysis for both cell types and observed a highly positive correlation (R = 0.089, *p* < 0.05; Figure 3B), indicating a high similarity in their gene expression patterns. In total, 2682 genes were identified as differentially expressed genes (DEG) between hiPSC-LEPC and T-LEPC (log2 fold change > 2.0; *p*-value < 0.05). 

Genes related to pluripotency, limbal stem cell and embryonic and adult corneal epithelial markers were analyzed [43]. RNA-seq analysis confirmed that pluripotent stem cell-related markers (Figure 4B(i)) showed either no difference in expression (MYC, KLF4, TBX3, SALL4, MYT1L, ASCL1) or lower expression (PRDM1; log2 fold-change = 4.9) in hiPSC-LEPC compared to T-LEPC, whereas SOX2, another pluripotent stem cell marker, was highly expressed in hiPSC-LEPC compared to T-LEPC while the expression levels were very low (0,T-LEPC vs. 61.7, hiPSC-LEPC normalized reads; Heatmap; Figure 3D). RNA-seq analysis revealed that both cell types showed similar expression patterns for the ocular developmental marker PAX6, LEPC markers (KRT19 (CK19), CDH3 (P-cadherin), KRT15 (CK15), TP63 (p63)) and markers of mature cornea (KRT12 (CK12), CDH1 (E-cadherin), GJA1, ITGB1, and ITGA3) (Figure 3C), strongly suggesting their differentiation into an LEPC phenotype. RNA-seq data confirmed a lower expression of KRT14 (CK14, a limbal/corneal basal epithelial marker; log2 fold-change = 3.6, padj = 0.0001) and a higher expression of CDH2 (N-cadherin, LEPC marker; log2 fold-change = 5.0, padj = 3.3 × 10^−6^), IVL (log2 fold-change = 2.6-fold), KRT3 (CK3; log2 fold-change = 5.7-fold; padj = 0.004, ALDH3A1 (log2 fold-change = 5.7-fold, padj = 0.00001), clusterin (CLU, log2 fold-change = 4.3-fold, padj = 2.0 × 10^−17^, corneal differentiation markers) in hiPSC-LEPC compared to T-LEPC (Figure 3C). hiPSC-LEPC also displayed higher levels of mucosal epithelial markers KRT13 and KRT4, whereas the membrane mucin marker MUC4, the unicellular gland marker KRT7, and the goblet cell marker MUC5AC were not different (Appendix A). There was no difference between hiPSC-LEPC and T-LEPC in epidermal marker KRT10 expression (Appendix A). Further, out of the five major oncogenes that are overexpressed in hiPSC [44], RAC3, RAB25, PIM2, and MYBL2 did not differ between hiPSC-LEPC and T-LEPC, whereas TET1 is slightly overexpressed in hiPSC-LEPC compared to T-LEPC (Appendix A).

To gain more insight into the biological pathways associated with DEGs, we performed a GO enrichment analysis of DEG in hiPSC-LEPC (1985) and T-LEPC (697), which showed that DEGs were highly involved in extracellular structure and matrix organization (bar plots, Figure 3C). The top 20 of highly expressed ECM-related T-LEPC genes includes LAMC2, LAMB3, LAMA3, CD44, SERPINE1, COL7A1 (Figure 3E(i)), which were expressed on the entire ocular surface, whereas hiPSC profile includes include SPARC, FN1, COL3A1, COL4A2, COL4A2, TNC, and versican (Figure 3E(ii)), which are highly expressed in the limbal niche [24,45,46]. 

### 3.3. Immunogenic Potential of hiPSC-LEPC

The potential role of hiPSC-LEPC in inflammation and immune regulation was studied in comparison to T-LEPC. Using gene set enrichment analysis (GSEA), we observed immune-related gene signatures (GO:0002821) enriched in T-LEPC compared with hiPSC-LEPC (Enrichment score (ES) = 0.4825; *p* < 0.05; Figure 4A). The heat map shows expression of immunorelated genes in hiPSC-LEPC in comparison to T-LEPC (Figure 4B). The genes encoding HLA-A (log2 fold-change = 3.0-fold, padj < 6.0 × 10^−15^), -B (log2 fold-change = 3.0-fold, padj = 2.3 × 10^−5^), -C (log2 fold-change = 3.1-fold, padj = 1.3 × 10^−25^) and co-stimulatory molecules CD40 (log2 fold-change = 4.5.0-fold, padj = 3.6 × 10^−15^), CD274 (PD-L1; log2 fold-change = 3.2-fold, padj = 3.1 × 10^−4^) were expressed more highly in T-LEPC compared to hiPSC-LEPC, whereas intercellular cell adhesion molecule 1 (ICAM1; log2 fold-change = 2.6-fold, padj = 3.5 × 10^−5^) was more highly expressed in hiPSC-LEPC (Figure 4B; Appendix A). The other co-stimulatory molecule (CD40L) and co-stimulatory molecule ligands (CD80 and CD86) were expressed at very low levels and did not show any difference between the samples (Figure 4B). 

According to the RNA sequencing data, the expression of major histocompatibility complexes MHC class I (HLA-ABC) and class II (HLA-DR), an immunosuppressive molecule (PD1), co-stimulatory molecules (CD40, CD40L) and their ligands (CD80, CD86) were investigated by flow cytometry. Almost all hiPSC-LEPC and T-LEPC expressed HLA-ABC (Figure 4C), but the median fluorescence intensity (MFI) of hiPSC-LEPC was lower compared to T-LEPC (Appendix A). After IFN-γ stimulation, MFI increased in both hiPSC-LEPC and T-LEPC without significant differences between the cell types (Appendix A). HLA-DR expression significantly increased in hiPSC-LEPC (0.7 ± 0.4 vs. 78.0 ± 6.1%; *p* = 0.02) and T-LEPC (0.4 ± 0.3 vs. 18.9 ± 2.8%; *p* = 0.02) after IFN-γ treatment and a significantly higher number of cells was positive for HLA-DR in hiPSC-LEPC than T-LEPC (78.0 ± 6.1% vs. 18.9 ± 2.8%; *p* = 0.02, Figure 4C). Fewer T-LEPC expressed CD40 than hiPSC-LEPC (10.0 ± 7.0 vs. 1.3 ± 0.6%; *p* = 0.02), but no expression was observed for other co-stimulatory molecules or their ligands (CD80, CD86, CD40L) in both cell types. The IFN-γ stimulation did not change the expression of CD40, CD40L, CD80, and CD86. ICAM-1 expression was detected in a significantly larger fraction of hiPSC-LEPC than T-LEPC (58.6 ± 19.5 vs. 20.7 ± 7.7%; *p* = 0.02), whereas IFN-γ treatment induced ICAM-1 in both cell types in more than 95% of cells (Figure 4C). The programmed cell death protein 1 (PD-1) was scarce at baseline in both LEPC types (1.1 ± 0.5 vs. 3.0 ± 3.7%; n.s.), and IFN-γ induced PD-1 more strongly in hiPSC than in T-LEPC (59.6 ± 28.7 vs. 12.6 ± 6.0%, *p* = 0.02). 

A mixed lymphocyte reaction (MLR) assay was used to investigate hiPSC-induced immune responses. LEPC of both origins were either left unstimulated or received IFN-γ treatment for 24 h prior to co-culture with PBMCs, which were either nonactivated or activated by anti-CD2/3/28 antibodies. PBMC proliferation was assessed by flow cytometry analysis of CFSE dilution after 72 h of co-culture. Independent of pretreatment, LEPC of both origins did not induce proliferation of nonactivated PBMCs (Figure 4D(i,ii)). Proliferation of activated PBMC was significantly inhibited by hiPSC-lEPC at high density (1/1), but unaffected by T-LEPC (26.4 ± 4.7 vs. 67.4 ± 30.0; *p* = 0.004; Figure 4D(iii)).

### 3.4. Angiogenic Potential of hiPSC-LEPC

We explored whether hiPSC-LEPC could stimulate angiogenesis or inhibit it compared to T-LEPC. Based on GSEA, hiPSC-LEPCs were enriched in angiogenesis-related gene sets in comparison to T-LEPCs (ES = −0.4; *p* < 0.05, Figure 5A). Hierarchical clustering analysis shows top 25 genes associated with angiogenesis (GO: 0001525) in hiPSC-LEPC and T-LEPC. Based on the significance of GO term enrichment, we focused on the biological functions “positive regulation of angiogenesis” (GO: 0045766) and “negative regulation of angiogenesis” (GO: 0016525) for a detailed analysis. A comparison of the respective read counts revealed that 16 of 30 genes associated with positive regulation of angiogenesis and 17 of 20 genes associated with negative regulation of angiogenesis were more strongly expressed in hiPSC-LEPC (Figure 5C(i,ii)). The dominance of genes involved in negative regulation of angiogenesis in hiPSC-LEPC suggest that hiPSC-LEPC supports the endothelial quiescence and appears compatible with an antiangiogenic effect of hiPSC-LEPC [47]. 

Next, we examined the effect of the hiPSC-LEPC secretome on the proliferation, migration, and three-dimensional sprouting of HUVECs in comparison to T-LEPC secretome. To compare the angiogenic potential of hiPSC-LEPC and T-LEPC, the respective conditioned media were examined in assays of three-dimensional spheroid sprouting, vascular endothelial cell proliferation and migration. Adding the pro-angiogenic growth factor vascular endothelial growth factor to the EBM significantly increased spheroid sprouting over fivefold (*p* < 0.0001, Figure 5D). HUVECs demonstrated significantly more sprouting in T-LEPC CM (undiluted) compared to hiPSC-LEPC CM (undiluted) (1.3-fold, *p* = 0.0008; Figure 5D). When CM was diluted (1:1) with EBM to reduce a possible nutrient depletion, no significant difference in sprouting was observed between the conditioned media (Figure 5D). Endothelial cell migration as assessed by a standardized scratch wound assay indicated that T-LEPC CM provides a better healing capacity compared to hiPSC-LEPC CM in undiluted samples (Figure 5E), whereas diluted CM did not show any differences (Figure 5E). The effects of CM on endothelial cell proliferation were assessed by BrdU incorporation assays 72 h after seeding of HUVECs. Compared with T-LEPC CM, hiPSC-LEPC CM significantly inhibited the proliferation of HUVECs (undiluted: 3.1-fold; *p* < 0.0001 and diluted: 2.0-fold, *p* < 0.0001) (Figure 5F). 

### 3.5. Repopulation of Decellularized Scaffolds

The repopulation potential of hiPSC-LEPC was tested by seeding these cells on DHC/L scaffolds. In addition, we used LM (CD90^−^CD117^+^P-cadherin^+^) as they share localization with LEPC in vivo. The seeded hiPSC-LEPC and LM attached to the surface of decellularized scaffolds and remained viable after 48 h of cultivation as confirmed by live/dead staining (Appendix A). After 10 days of cultivation, stratification (2–3 layers in the cornea; 5–6 layers at limbus) of the epithelium was observed in H&E-stained tissue sections (the dotted line represents the BM; Figure 6A). 

Phenotypic characterization of repopulated limbal scaffolds by immunohistochemical staining confirmed pronounced epithelial keratin (pan-CK) expression and intercellular E-cadherin in all epithelial layers; a proliferation marker Ki-67 expression in the basal cells; p63, CK15 and CK17/19 (arrowheads) were detected in the basal layer; CK3/76 and CK12 expression in the suprabasal cells; Melan-A^+^ melanocytes were interspersed in the epithelial layers (arrowhead), (dashed line represents BM; Figure 6B). We also investigated BM components in recellularized scaffolds and found Col IV, Fibronectin and LN-α3 largely intact (Appendix A). 

The TEM analyses of tissue-engineered hiPSC-LEPC constructs showed 3 to 4 layers of cells (Figure 6C(i)), keratin filaments in superficial cells and more pronounced desmosomes (Figure 6C(ii)). The cells appear loosely connected to each other and to the underlying stroma (Figure 6C(i,ii)). Tissue-engineered hiPSC-LEPC constructs co-cultured with LM showed a 5–7 layered epithelium, loosely connected, melanocytes between the epithelial cells and melanosome-like structures around the nuclei (Figure 6C(iii,iv)). Of note, a basal deposition of abnormal ECM material (Figure 6C(v)), including microfibrils and elastic fibrils was observed (Figure 6C(vi)). We also looked at RNA-seq data of hiPSC-LEPC and T-LEPC for the expression of ECM-related genes, which are involved in abnormal deposition of ECM in pathological conditions. We observed the significant upregulation of ECM-related genes (FBN1, LTBP1, EFEMP2, TIMP2, CLU, LOXL1, LOXL4 and ELN) in hiPSC-LEPC compared to T-LEPC (Figure 6C(vii)). These data suggest that the cultivation of hiPSC-LEPCs on decellularized scaffolds promotes the generation of multilayered stratified corneal epithelial tissue equivalents. However, the abnormal ECM production may pose challenges to clinical use and requires further elucidation.

## 4. Discussion

Transplantations of allogeneic LEPCs or autologous non-ocular cells are the only current options for treating bilateral LSCD [43,48]. Unfortunately, the clinical success of allogenic LEPC transplantation is still hampered by immune rejection and autologous ocular mucosal cells have only a weak anti-angiogenic potential [44,45]. In light of these limitations, transplantation of recipient-derived or HLA-matched hiPSC-LEPC holds considerable potential as a treatment strategy for LSCD, and a detailed functional characterization of hiPSC-LEPC is critical. Here, we extensively compared T-LEPCs and LEPCs generated from hiPSC via a SEAM method reported by Hayashi et al. [13,42]. Our data show that hiPSC-derived CD200^−^ITGB4^+^SSEA4^+^ cells have similar wound healing capacity, melanosome uptake, the ability to form colonies and expression of LEPC markers (CK15, CK19, P-cadherin, CK14, and p63) as T-LEPC [14,42,46]. Our data strongly suggest that hiPSC-LEPC are more similar to progenitor and transient amplifying cells than terminally differentiated corneal epithelium and their potential to regenerate the corneal epithelium is similar to the capacity observed in vivo [46,49,50,51]. 

The success rate of hiPSC-LEPC therapy is highly dependent on the differentiation potential of the transplanted cells, including their ability to home to the limbus and function as stem cells. To further investigate the similarities between hiPSC-LEPC and T-LEPC, gene expression was assessed using RNA-sequencing. Several candidate LEPC markers such as TP63, KRT15, CK19, CDH3, and MSX2, were similarly expressed in hiPSC-LEPC and T-LEPC. Markers for terminally differentiated corneal epithelium such as KRT3, KRT12, and IVL showed very low expression similar to earlier reports [52]. However, bulk RNA-seq data also indicated a higher expression of KRT13 in hiPSC-LEPC compared to T-LEPC, suggesting the presence of some non-LEPCs (maybe conjunctival epithelial cells) in cultures as reported earlier [42,53]. However, other genes related to conjunctival cells like MUC4 and MUC5AC were completely absent, whereas KRT19 and K7 did not show differential expression between the hiPSC-LEPC and T-LEPC. Moreover, CK13 expression was also reported in suprabasal limbal epithelium both at mRNA [54] and protein levels [55,56]. These data suggest that better purification techniques are required to isolate a purer population of hiPSC-LEPC. In both hiPSC-LEPCs and T-LEPCs, epidermal marker KRT10 expression was similar, suggesting that KRT10 is expressed at a basal level across native epithelial and iPSC-derived epithelial cells. 

Immunogenicity of hiPSC-LEPC would also be critical for successful transplantation and the long-term survival of the transplanted cells. It has been shown that an inflammatory microenvironment increases MHC molecule expression and enhances leukocyte recognition [57]. Additionally, it increases the infiltration of activated immune cells, alters the function of immunomodulatory cells, and affects immunogenicity, function, and survival of donor cells. In general, derivatives of embryonic stem cells and iPSC were shown to express less HLA-1 and to be less antigenic than somatic cells [58,59]. As reported previously, the current study also found that a broad range of immune-related genes were downregulated in hiPSC-LEPCs, including HLA-1 and CD40. It has been reported that differentiated corneal epithelial cells from ESC or iPSC showed weaker HLA-II expression than native corneal epithelial cells [16,17]. In contrast, our study showed significant IFN-γ-induced upregulation of HLA-II in hiPSC-LEPC compared to T-LEPC. The differences in findings may be due to the different control groups used i.e., P-cadherin^+^ cells versus unsorted LEPC. However, further studies are necessary to validate these observations. We have previously reported that inflammatory conditions are associated with limbal infiltration of CD45^+^ leukocytes and CD11c^+^ dendritic cells that interact directly with LEPC via ICAM-1 [24]. In line with T-LEPC, hiPSC-LEPC showed moderate expression of ICAM-1 under homeostatic conditions whereas activation of hiPSC-LEPC with inflammatory stimuli (IFN-γ) increased their surface expression of ICAM-1 by over 90% [24]. It has been reported that PD1 is required for prolonged corneal allograft survival and plays a critical role in corneal immune privilege [60,61,62]. The surface expression of PD1 increased in T-LEPC and hiPSC-LEPC in response to inflammatory stimuli, and the upregulation of PD1 was significantly higher in hiPSC-LEPC than in T-LEPC, suggesting that hiPSC-LEPC exhibit immunomodulatory properties. In vitro, corneal epithelial cells derived from ESC or iPSCs were found to be less immunogenic and responsible for less proliferation of T cells and PBMCs, respectively [16,17]. In the current study, activated or not activated hiPSC-LEPC and T-LEPC did not show any effect on the proliferation of PBMCs, whereas hiPSC-LEPCs inhibited the proliferation of activated PBMCs. These observations suggest low immunogenicity of hiPSC-LEPCs in vitro, but further research is needed to better understand the mechanisms and immunomodulatory functions of these cells.

Corneal avascularity is important for optical transparency and mainly depends on the barrier function of the limbal niche, which produces balanced pro- and anti-angiogenic factors during homeostasis [63]. In LSCD, the loss of LEPC or niche factors (ECM and limbal niche cells) results in corneal neovascularization across the limbal barrier [2]. Therefore, current LSCD treatment strategies aim at not only restoring damaged corneal epithelium but also minimizing corneal neovascularization. It has been reported, that eyes transplanted with cultured oral mucosal epithelial cell sheets have shown increased neovascularization compared to eyes treated with LEPC [7,45]. As a result, the angiogenic potential of hiPSC-LEPC is a critical matter to be resolved before clinical application. According to RNA-seq data, hiPSC-LEPC maintain both proangiogenic and antiangiogenic regulators, suggesting the potential to maintain an avascular state similar to T-LEPC [63]. In vitro studies have shown that LEPC did not influence proliferation, migration, or tube formation of EC [26] but inhibited vascular endothelial cell morphogenesis [64]. Similar to these studies, in the current study, the secretome of hiPSC-LEPC showed a stronger anti-proliferative effect on endothelial cells than the T-LEPC secretome. Even though we observed a significant difference in undiluted samples of hiPSC-LEPC and T-LEPC in spheroid and migration assays, the effect was completely nullified when diluted. This might be due to released factors or to the depletion of nutrients in undiluted CM of hiPSC-LEPC. These observations suggest that hiPSC-LEPC have a stronger anti-angiogenic effect than T-LEPC and might reduce corneal neovascularization in LSCD treatment.

Various biomaterials, mainly amniotic membranes (AM) and fibrin gels have been used to cultivate LEPC for treating LSCD [4,5]. Biological thickness variability as well as poorly standardized techniques are the major limiting factors of AM, which caused a significant variation in clinical outcomes [65,66]. Decellularized scaffolds have a unique advantage of having native ECM composition and three-dimensional architecture [67,68]. We have reported earlier that decellularized limbal scaffolds provide a limbus-specific microenvironment for the expansion of LEPC as a promising scaffold for the treatment of LSCD [23]. The current study demonstrated the efficient recellularization of human corneolimbal scaffolds in vitro as indicated by a stratified epithelium with expression of limbal epithelial markers and the presence of melanocytes in the epithelial layers of DHL-hiPSC-LEPC-LM scaffolds. DHL scaffolds repopulated with hiPSC-LEPC may eventually be applicable for transplantation in LSCD patients. However, in vivo studies are necessary for determining immunogenicity and in vivo biocompatibility. On the other hand, hiPSC-LEPC showed an upregulation of ECM-related genes and the deposition of fibrillar and abnormal ECM in recellularized scaffolds. These observations could indicate that cells might undergo abnormal differentiation in prolonged culture and represent a pseudoexfoliation-like pathological condition. The deposited material depicted by electron microscopy shows morphological similarity to ECM changes observed with pseudoexfoliation syndrome, an alteration of ECM processing which is associated with deposition of flocculent protein material in the anterior chamber of the eye [69]. It is currently unclear whether the deposits observed with iPSC-derived LEPC are associated with changes in LOXL1 function as has been suggested for pseudoexfoliation syndrome [70]. Brezesczynska J et al., [71] reported that long-term cultured corneal epithelial cells derived from ESC underwent transdifferentiation to a phenotype of squamous metaplasia. However, other iPSC cell lines must be tested before any additional conclusions can be drawn as the results of our study are limited to one iPSC line. Therefore, further studies are needed to explore this issue, and reconstructed tissue-engineered limbal scaffolds containing hiPSC-LEPC and LM need to be tested in appropriate LSCD animal models. 

In conclusion, hiPSC-LEPCs showed functional properties similar to native LEPC regarding colony formation, melanosome uptake, and healing ability. Transcriptomic analysis also revealed a high similarity between hiPSC-LEPC and T-LEPC. The hiPSC-LEPC also appeared less immunogenic and angiogenic compared to T-LEPC. Finally, hiPSC-LEPC successfully repopulated limbal scaffolds. As a caveat, the present study also revealed a previously unreported challenge, i.e., an abnormal ECM deposition by hiPSC-LEPC, which needs to be resolved prior to clinical application.

## Figures and Tables

**Figure 1 cells-11-03752-f001:**
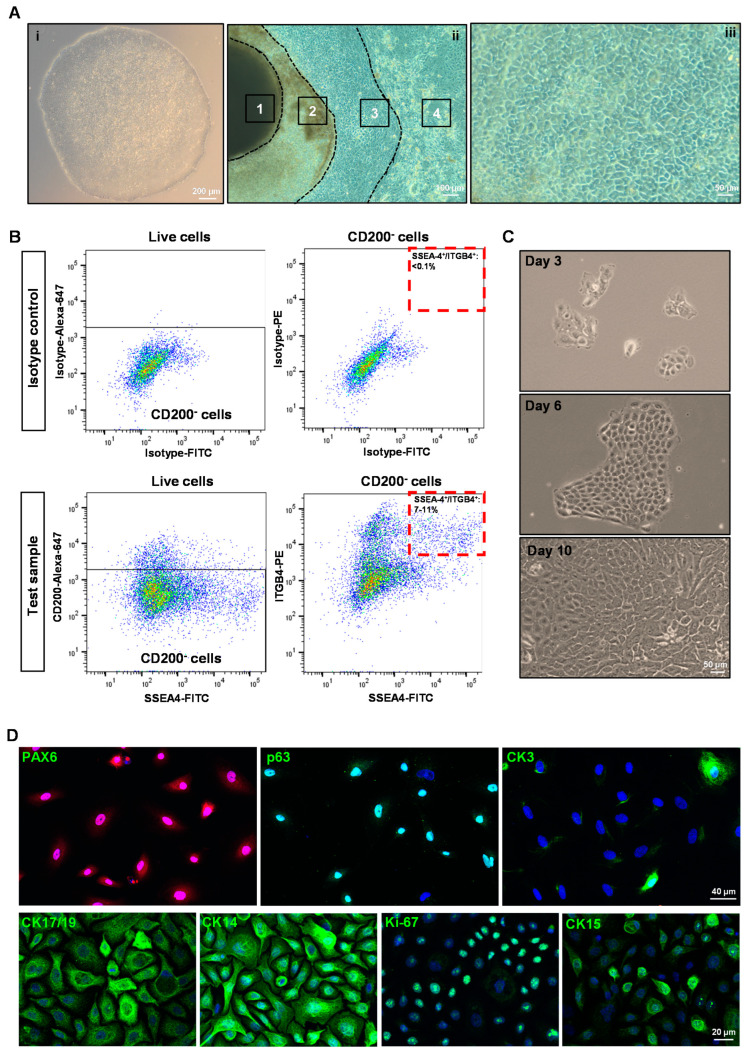
Isolation and characterization of human-induced pluripotent stem cell-derived limbal epithelial progenitor cells (hiPSC-LEPC): (**A**) phase contrast micrographs show the colony of hiPSC (**i**), a typical self-formed ectodermal autonomous multi-zone (1-neuroectoderm; 2-optic cup and neural crest; 3–ocular surface ectoderm; 4-surface ectoderm) of hiPSC cell after six weeks of differentiation (**ii**) and differentiated epithelial cells after 12 w of differentiation (**iii**); (**B**) fluorescence-activated cell sorting images demonstrate the gating strategy of isolating hiPSC-LEPC. The isotype control graphs (upper panel) showing the set of gates to select CD200^−^ (upper left) cells initially and later for CD90^−^SSEA4^+^ITGB4^+^ cells (Upper right). The test sample graphs show the selection of CD200^−^ cells (lower left) and CD200^−^SSE4^+^ITGB4^+^ cells (lower right). Data are expressed as the percentage means ± standard error of the mean of 13 independent cell-sorting experiments; (**C**) phase contrast micrographs show cultured hiPSC-LEPC exhibited cuboidal epithelial phenotype and grew like colonies; and (**D**) immunostaining of cultured hiPSC-LEPC (P1) showing the expression PAX6 (in all cells), p63, CK14, CK19 (>95%), and CK15 (30–40%), CK3 (~5 to 10%) and the proliferation marker Ki-67 (~50–60% cells). Nuclear counterstaining with 4′,6-diamidino-2-henylindole (blue).

**Figure 2 cells-11-03752-f002:**
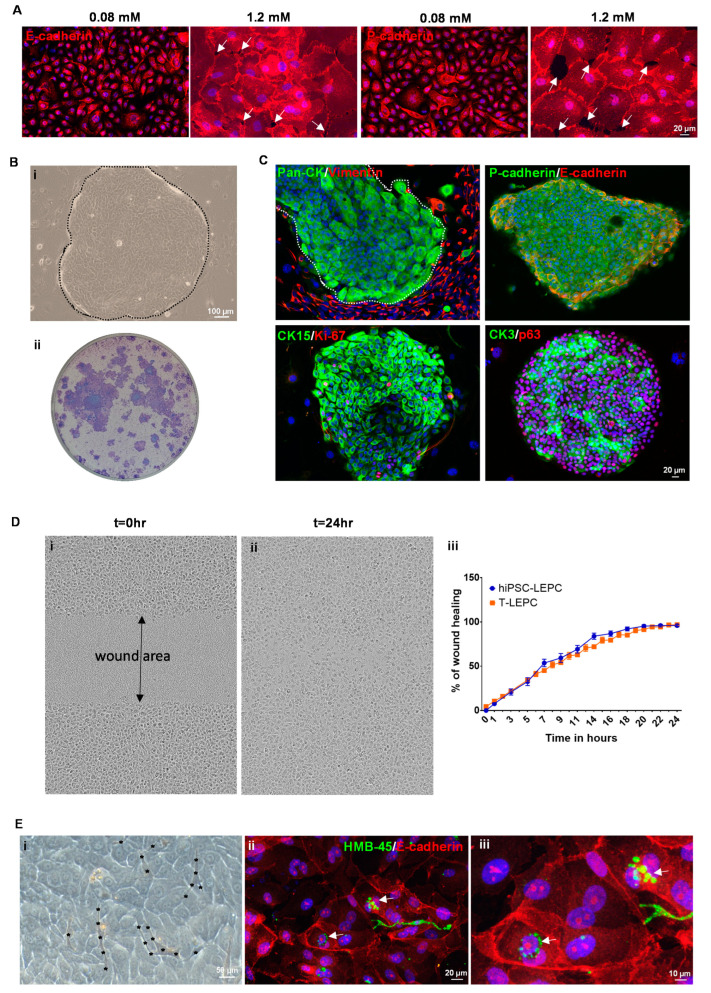
Functional characterization of hiPSC-LEPC: (**A**) immunostaining of cultured hiPSC-LEPC showing diffuse staining of E-cadherin and P-cadherin in 0.08 mM Ca^2+^ and enriched staining at cell-cell junctions in 1.2 mM Ca^2+^. White arrows showing the gaps between the cells. Nuclear counterstaining with 4′,6-diamidino-2-phenylindole (DAPI, blue); (**B**) phase contrast micrograph showing the hiPSC-LEPC colony (dotted circle) with smooth borders on the NIH/3T3 fibroblast feeder layer (**i**); stained colonies with crystal violet (**ii**); (**C**) double immunostaining of hiPSC-LEPC colonies on the NIH/3T3 fibroblast feeder layer show the membranous staining of E- and P-cadherin; limbal epithelial progenitor marker CK15 (green) and the proliferation marker Ki-67 (red) mostly at the edges of the colonies; and corneal differentiation maker CK3 (green) and the progenitor cell maker p63 (red) in almost all cells of the clone. Nuclear counterstaining with DAPI (blue); (**D**) the phase contrast micrograph of the hiPSC-LEPC shows the wound formation (t = 0, (**i**)) and healing of a wound after 24 h (t = 24, (**ii**)). The graph shows wound healing capacity of hiPSC-LEPC and T-LEPC at different time points (**iii**). Data are expressed as the percentage means ± standard error of the mean of four independent experiments; and (**E**) phase contrast micrograph showing the infiltration of limbal melanocytes (LM) in the hiPSC-LEPC colonies (stars depict the melanocytes and their processes, (**i**)). Double immunostaining of hiPSC-LEPC and LM co-cultures showing the accumulation of pigment globules (HMB45, green) in the perinuclear cytoplasm of hiPSC-LEPC (arrow, (**ii**,**iii**)). Nuclear counterstaining with DAPI (blue).

**Figure 3 cells-11-03752-f003:**
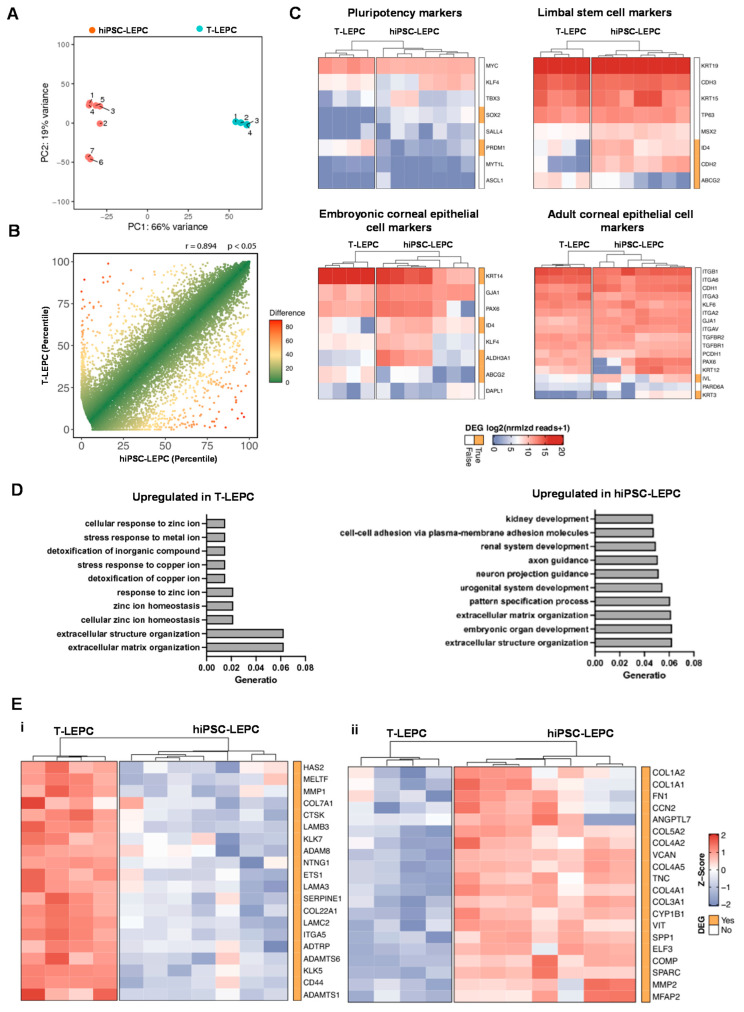
Transcriptional profile of hiPSC-LEPC: (**A**) principal component analysis (PCA) plot illustrating the distribution of hiPSC-LEPC (n = 7) and T-LEPC (n = 4) populations; (**B**) graphical presentation of the percentiles of normalized counts to illustrate similarities between the transcriptome of hiPSC-LEPC and T-LEPC. The expression of each gene in each cell population was calculated as a percentile (the gene with the highest number of normalized counts getting the percentile 100, the one with the lowest—the percentile 0). The higher the Pearson coefficient R, the higher the similarity between the hiPSC-LEPC and T-LEPC; (**C**) heatmaps showing the transcriptional profiles of genes related to pluripotency, limbal stem cells, and corneal epithelial markers coding for the log2 of normalized reads +1. Rows and columns are clustered according to the similarity of expression. The red color in the Heat map represents highly expressed genes, while blue represents low expression genes; The bar next to the heatmap represents similar (white) and differentially expressed genes (DEGs—log2 fold change > 2.0; *p* < 0.05; orange); and (**D**) GO enrichment analysis using the DEGs between hiPSC-LEPC and T-LEPC. The ten most enriched GO terms according to their adjusted *p* value are visualized. (**E**) The heatmaps show the top 20 of highly expressed DEGs related to extracellula matrix organization (GO:0030198) using the z-score in the T-LEPC (**i**) and hiPSC-LEPC (**ii**).

**Figure 4 cells-11-03752-f004:**
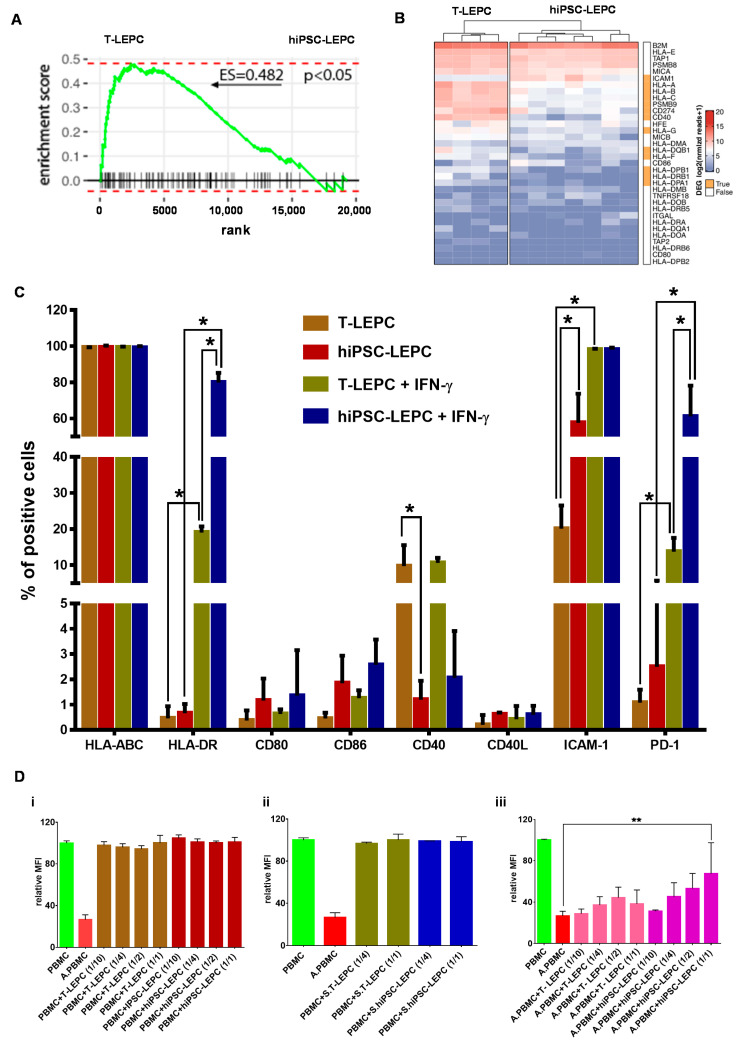
Human iPSC-LEPC in immune regulation: (**A**) enrichment plots showing the curves of GSEA enrichment score for immune-related gene set in LEPC vs. hiPSC-LEPC. The immune-related gene set was significantly enriched in T-LEPC; (**B**) the heatmap showing the log2 of normalized reads + 1 of immune-related genes. Rows and columns are clustered according to the similarity of expression. The red color in the heatmap represents highly expressed genes, while blue represents low expression genes; The bar next to the heatmap represents similar (white) and differentially expressed genes (DEGs—log2 fold change > 2.0; *p* < 0.05; orange); (**C**) flow cytometry analysis of hiPSC-LEPC and T-LEPC in the presence or absence of interferon-γ showing the expression of markers. Percentage of cells expressed as mean ± standard error of the mean (n = 4). * *p* < 0.05; and (**D**) flow cytometric analysis of peripheral blood mononuclear cells (PBMC) proliferation by CFSE intensity showing PBMC proliferataion was not affected by co-culturing with either unstimulated (**i**) or stimulated (**ii**) T-LEPC or hiPSC-LEPC, whereas the proliferation of activated PBMCs (A.PBMCs) with hiPSC-LEPC (1/1) was significantly reduced (**iii**). Data are expressed as mean ± standard error of the mean (n = 4). ** *p* < 0.01.

**Figure 5 cells-11-03752-f005:**
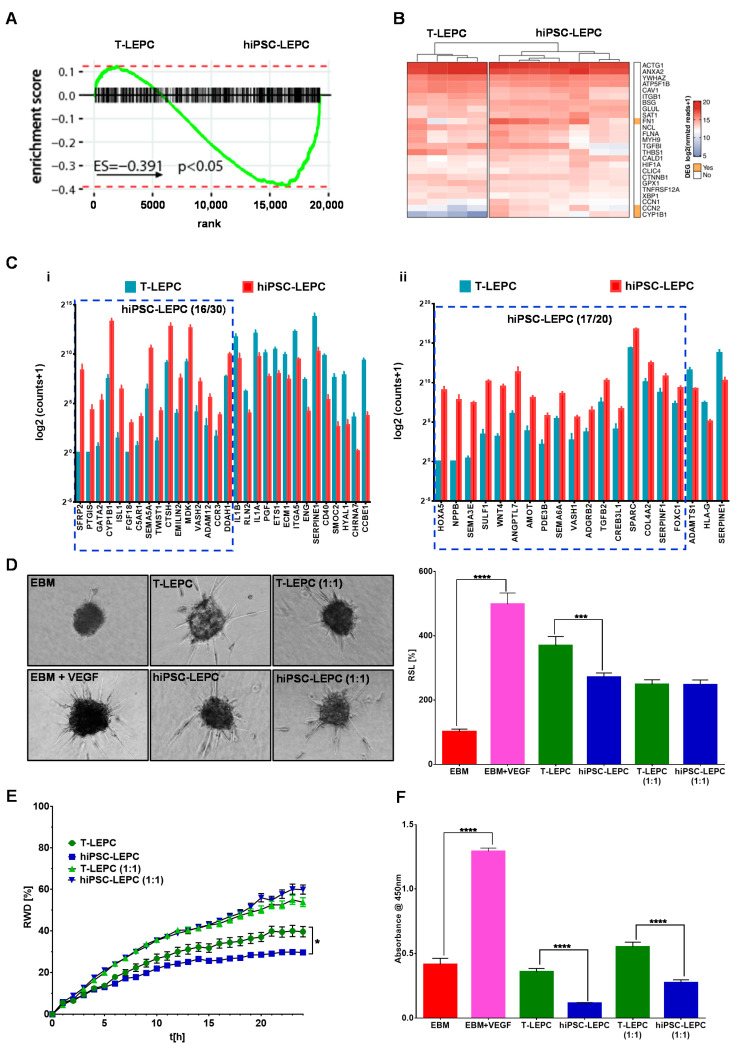
Human iPSC-LEPC in angiogenesis: (**A**) enrichment plots showing the curves of GSEA enrichment score for angiogenesis-related geneset (GO:0048514) in T-LEPC vs. hiPSC-LEPC. The angiogenic gene set was significantly enriched in hiPSC-LEPC; (**B**) the heatmap shows the log2 of normalized reads + 1 of the top 25 highly expressed angiogenesis-related genes in both hiPSC-LEPC and T-LEPC; (**C**) the graphs show the gene ontology of biological clusters of genes involved in positive regulation of angiogenesis (**i**, GO:0045766) and negative regulation of angiogenesis (**ii**, GO:0016525). All the genes illustrated in the graphs are significantly differentially expressed; (**D**) phase-contrast micrographs show the sprouting of vascular endothelial cells in different conditions. Quantification of sprout length relative (RSL) to EBM in percentage (%). Data are expressed as means ± standard error of the mean (n = 7). *** *p* < 0.001; **** *p* < 0.0001; (**E**) the line graph shows the relative wound density of endothelial cells over time in different conditions. Data are expressed as means ± standard error of the mean (n = 7). * *p* < 0.05; and (**F**) the effect of CM of T-LEPC and hiPSC-LEPC on endothelial cell proliferation was analyzed by BrdU incorporation after 72 h of incubation. Data are expressed as means ± standard error of the mean of 4 individual experiments. **** *p* < 0.0001 Mann–Whitney U test.

**Figure 6 cells-11-03752-f006:**
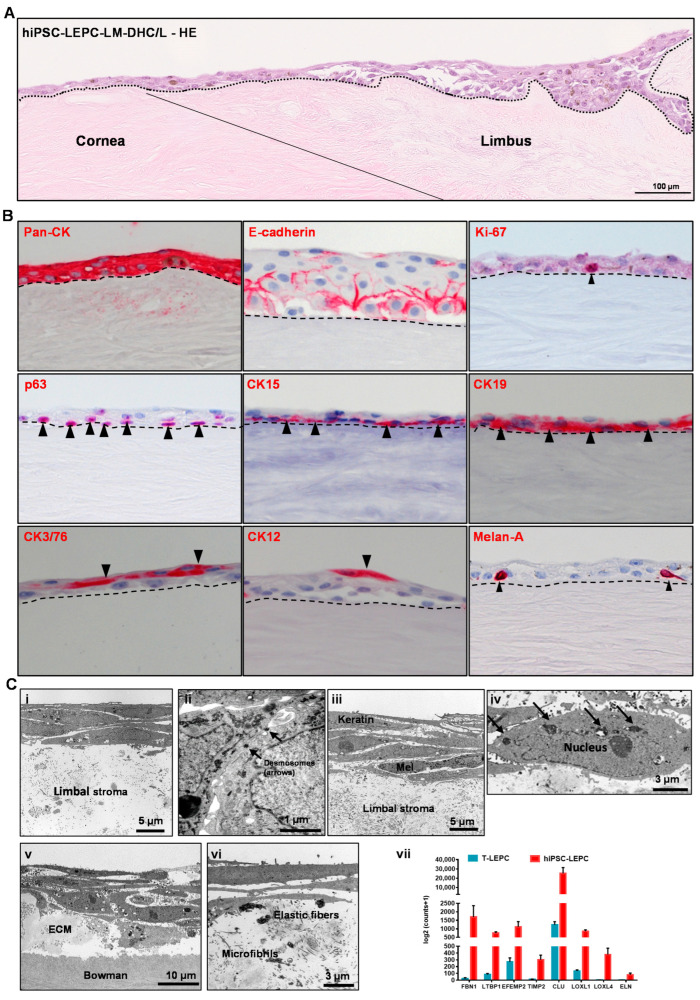
Recellularization of decellularized human corneolimbal (DHC/L) scaffolds: (**A**) hematoxylin and Eosin (HE) staining of recellularized scaffold showing the stratification (2–3 layers) of the epithelium both at the corneal and limbal region (the dotted line represents basement membrane (BM)); (**B**) immunohistochemical staining of recellularized corneal/limbal scaffolds showing pronounced epithelial keratins (pan-cytokeratin (pan-CK)) expression and intercellular E-cadherin in all epithelial layers; p63, CK15, and CK17/19 staining (arrowheads) in the basal layer; CK3/76 and CK12 staining in the superficial layers of the epithelium (arrow heads); Melan-A^+^ melanocytes interspersed in the epithelial layers (arrowhead). The dashed line represents BM; and (**C**) the electron micrographs show three to four layered stratified epithelium (**i**), more pronounced desmosomes (**ii**) in hiPSC-LEPC constructs; 5–7 multi-layered epithelium, presence of melanocytes (**iii**) between the epithelial cells, and melanosome-like structures around the nucleus (**iv**, arrows) in hiPSC-LEPC/LM constructs. The electron micrographs also show a basal deposition of pseudoexfoliation-related extracellular matrix material (**v**) including elastic fibers and microfibrils (**vi**). The RNA sequencing analysis (**vii**) of hiPSC-LEPC and T-LEPC illustrate the significantly higher expression of transcripts related to pseudoexfoliation in hiPSC-LEPC compared to T-LEPC (log2fold change > 2.0; *p* < 0.05). Data are expressed as means ± standard error of the mean.

## Data Availability

The datasets generated during and/or analyzed during the current study are available from the corresponding author on reasonable request.

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
