# Peer review of "Transcriptomic Landscape and Functional Characterization of Human Induced Pluripotent Stem Cell-Derived Limbal Epithelial Progenitor Cells"

_cells, 2022, doi:10.3390/cells11233752_

Round 1

Reviewer 1 Report

In the manuscript, authors represent data regarding characterization of hiPSC- LEPC generated using a previously established clinically applicable protocol. The phenotypic similarity of hiPSC-LEPC to native T-LEPC was investigated by transcriptomic analysis, immune regulation and angiogenesis assays. In addition, the ability of hiPSC-LEPC to repopulate DHC/L scaffolds was conducted and interesting findings regarding possible abnormal ECM production is suggested.

In general, this is interesting and well conducted study and this reviewer has a few revision suggestions.

In the introduction, authors cite many papers describing hiPSC differentiation into corneal epithelial cells/lineages but they use terminology “Human limbal stem/progenitor cells (LEPC)” for their own differentiated cells. Authors could shortly justify in the introduction the potential difference (if any) why they consider that LEPC terminology is important for their iPSC differentiated cells as compared to terminology used in the other published protocols (e.g. Hyashi et al).

Material/methods:

Authors are using only 1 iPSC line for cell differentiation and thus data is quite limited. Authors should at least discuss this as a potential study limitation due to the widely acknowledge differentiation variability between different iPSCs and/clones. To increase the quality of the study, additional iPSCs should be used to confirm at least important findings (e.g. abnormal ECM production).

Authors should also include details/number of cell batched (n=) used in each analyses.

Flow cytometry, please state that appropriate compensation was carried out along with the multicolor flow cytometry analysis.

Results and discussion:

In Figure 1D, quite few/low number of cells presented in representative IF stainings for PAX6, p63, CK3. Authors should provide numbers of cells counted as positive? In supplementary Figure S1 western blot analyses is conducted for 8 individual experiments but it seems that there is a high variability e.g. with PAX6 expression levels and thus data is very different from results received with IF staining?

Data indicates “a higher expression of KRT13 in hiPSC-LEPC compared to T-LEPC, suggesting the presence of some non-LEPCs (maybe conjunctival epithelial cells)”. Similarly, it seems that cells have similar expression of epidermal marker KRT10. Authors could shortly discuss this as well.

In discussion authors propose that “DHL scaffolds repopulated with hiPSC-LEPC may eventually be applicable for transplantation in LSCD patients”. This is interesting possibility but authors could shortly discuss some of the potential limitations e.g. shortage of donor tissues, problems of incomplete decellularization etc.

Authors propose interesting finding that hiPSC-LEPC may produce abnormal ECM in vitro (RNAseq data) and on recellularized scaffolds. It would be important that authors discuss this matter further as the data presentation and discussion regarding this part is very short (e.g. pseudoexfoliation).

Minor, some of the references are numbered and some with “last name, year” format e.g. Hyashi et al., 2016

Minor typo, Fig 3C sub-headline “corneal epithelal” in 3rd blot

Reviewer 2 Report

Comments to authors

Polisetti et al., characterized hiPSC-LEPC in relation to T-LEPC, at the phenotypic and at the whole transcriptome level. Authors used a well-established protocol for cell differentiation and performed bulk RNA sequencing for analyzing the transcriptome. The angiogenic potential of the differentiated cells and their ability to repopulate decellularized corneal/limbal rim was evaluated. Authors found out that hiPSC-LEPC were very similar to T-LEPC at the phenotypic and at the molecular level. It was also observed that conditioned media from cultured hiPSC-LEPC had anti-angiogenic effect on HUVECs. Taken together, it was found that hiPSC-LEPC were very similar to native LEPC and that hiPSC-LEPC DHC/L scaffolds could be feasible for transplantation in patients suffering from LSCD in the future.

In general, this is very well performed study. The methods are clearly written and easy to follow. The results are properly, and clearly presented, and the discussion is within the scope of the study. I think it is a great piece of work.

The following comments are meant to clarify a few points mainly in the introduction

1.      The following statement in the introduction needs to be revised for clarity. Lines 64 to 67. ‘The gene expression profile of hiPSC-derived corneal epithelial cells has been analyzed in comparison with native corneal epithelial cells (paired box (PAX)6+ cytokeratin(CK)12+) 13 but not with P-cadherin+CD90-CD117- T-LEPC14 , which are basal cells of the limbal epithelium (PAX6+CK12- )’. 

2.      In the introduction lines 71 to 73, authors write about ‘low immunogenicity’ of other cell types. I wonder what the baseline is for this? Could the baseline being referred to be the level of immunogenicity of native LSC, or that of CEP? It would be great to clarify this, for better context. 

3.      In the results figure 2 D, it would be great if the cells can be stained with Calcein AM prior to imaging, to enhance visualization. This will make visualizing the gap a lot easier.
